# A Novel Approach for Effective Multi-View Clustering with Information-Theoretic Perspective

**Chenhang Cui**[1]*, **Yazhou Ren**[1]*†, **Jingyu Pu**[1], **Jiawei Li**[1],
**Xiaorong Pu**[1], **Tianyi Wu**[1], **Yutao Shi**[1], **Lifang He**[2]

[1] University of Electronic Science and Technology of China, China

[2] Department of Computer Science and Engineering, Lehigh University, USA

chenhangcui@gmail.com, yazhou.ren@uestc.edu.cn, pujingyu0105@163.com,
puxiaor@uestc.edu.cn, sixteen.l.jw@gmail.com, syt_59421@outlook.com,
tianyi-wu@outlook.com, lih319@lehigh.edu

## Abstract

Multi-view clustering (MVC) is a popular technique for improving clustering performance using various data sources. However, existing methods primarily focus on acquiring consistent information while often neglecting the issue of redundancy across multiple views. This study presents a new approach called sufficient multi-view clustering (SUMVC) that examines the multi-view clustering framework from an information-theoretic standpoint. Our proposed method consists of two parts. Firstly, we develop a simple and reliable multi-view clustering method SCMVC (simple consistent multi-view clustering) that employs variational analysis to generate consistent information. Secondly, we propose a sufficient representation lower bound to enhance consistent information and minimise unnecessary information among views. The proposed SUMVC method offers a promising solution to the problem of multi-view clustering and provides a new perspective for analyzing multi-view data. To verify the effectiveness of our model, we conducted a theoretical analysis based on the Bayes Error Rate, and experiments on multiple multi-view datasets demonstrate the superior performance of SUMVC.

## 1 Introduction

Clustering is a fundamental unsupervised learning task with broad applications across various fields. Traditional clustering methods primarily focus on analyzing single-view data. However, the rapid progress in multimedia technology has led to the collection of real-world data from multiple sources and perspectives, resulting in the emergence of multi-view data. Recently, multi-view clustering has garnered significant attention due to its ability to leverage information from multiple views to improve clustering performance and robustness [24]. Multi-view data arise in many applications where multiple sources of information or perspectives on the same set of objects are available. Multi-view clustering aims to group these objects into clusters by considering all available views jointly. This technique has proven to be highly effective in various domains, including social network analysis [6], image and video analysis [50], and bioinformatics [14]. Nevertheless, the main challenge in multi-view clustering lies in extracting valuable information from the views for clustering purposes while simultaneously minimizing view redundancy.

Information bottleneck (IB) theory provides a theoretical framework for understanding the optimal trade-off between accuracy and compression in data processing [36]. The IB method suggests that

---

*These authors contributed equally.

†Corresponding author.

37th Conference on Neural Information Processing Systems (NeurIPS 2023).

the optimal representation of the data can be achieved by maximizing the mutual information between the input and the output while minimizing the mutual information between the input and the representation. Although this principle provides a theoretic method to learn less superfluous information for downstream tasks, existing methods can still be improved in efficiently extracting embedded features from multi-view unlabeled data. In recent years, many information-theoretic methods have been applied to multi-view scenarios. Specifically, [44] proposes a generic pretext-aware residual relaxation technique to enhance multi-view representation learning (MVRL). [15] propose a novel approach to MVRL through an information-theoretic framework grounded in total correlation. Meanwhile, variational analysis has gained wide usage in machine learning and optimization for approximating complex probability distributions [3]. This technique offers a robust framework for solving optimization problems involving probability distributions, which are frequently encountered in multi-view clustering and information bottleneck. By approximating the probability distributions, variational analysis can effectively reduce the data dimensionality and enhance the efficiency of clustering algorithms. Variational autoencoders (VAEs) are a popular type of generative models that uses variational inference to learn compact representations of high-dimensional data [18]. However, the vast majority of them are designed for single-view data and cannot capture consistent and complementary information in a multi-view setting.

To tackle the aforementioned concerns, we delve deeper into the principles of the information bottleneck and variational analysis in the realm of multi-view clustering. We propose a simple consistent multi-view clustering method SCMVC and a novel multi-view self-supervised method SUMVC from an information-theoretic standpoint. SUMVC could strengthen consistency across views for clustering while concurrently decreasing view redundancy.

Our contributions are three-fold:

- We propose a consistent variational lower bound to explore the consistent information among views for multi-view clustering. We introduce SCMVC, a simple consistent multi-view clustering approach, utilizing the proposed lower bound. Despite being a simple approach, our experiments demonstrate the effectiveness of SCMVC.

- We extend the information bottleneck principle to the multi-view self-supervised learning and propose a sufficient representation lower bound. We expand SCMVC to improve the consistency and reduce the redundancy. This mainly aims to acquire predictive information and reduce redundancy among views. We refer to this new model as SUMVC (sufficient multi-view clustering).

- We adopt the collaborative training strategy for SUMVC to better adapt to multi-view scenarios. Additionally, a theoretical analysis of the generalization error of representations is conducted to evaluate the efficacy of SUMVC for clustering tasks. Experiments on real-world multi-view data show the superior performance of the proposed model.

## 2 Related Work

**Information Bottleneck** The information bottleneck (IB) principle is a widely used framework in information theory and machine learning that provides a rational way to understand the balance between compression and prediction. Introduced by [36], the IB principle formalises the intuition that successful learning algorithms need to find a balance between retaining relevant information and discarding irrelevant information. Since then, the IB principle has been widely applied to various machine learning tasks, including unsupervised learning [37], classification [3], and clustering [2, 51]. Recently, the IB principle has also been used to study the information processing capacity of the brain [33, 34]. By constraining the learned representations of deep neural networks, the IB principle was shown to improve the generalization performance of these models and reduce their computational complexity [2]. In addition, the IB principle has been used to develop new techniques for training deep neural networks, such as the deep variational information bottleneck [2]. The theoretical foundation of the IB principle is built on concepts such as Markov chains, entropy, and conditional entropy [5]. It has been widely applied in various fields, such as data mining, image processing, natural language processing, and computer vision [10, 16, 35]. The IB principle has also been applied in control theory to quantify the amount of [31]. Overall, the IB principle has become an important tool for designing more efficient and effective machine learning algorithms. Its widespread applications in various fields highlight its relevance and practicality.

**Varitional Autoencoder** Variational autoencoders (VAEs) are a popular generative model class that learns a compact representation of input data through an encoder-decoder architecture. Unlike traditional autoencoders, VAEs are designed to model the latent space distribution of input data, enabling them to generate new data samples consistent with the input distribution. Early work on VAEs by [18] introduced the VAE model and derived its objective function, while [19] proposed a method for approximating gradients by sampling the latent variables. Since then, VAEs have been applied to various applications, including image generation, data compression, and anomaly detection. For instance, [11] combined VAEs with generative adversarial networks (GANs) to generate high-quality images, while [25] applied VAEs to detect anomalies in seasonal key performance indicators (KPIs). Moreover, VAEs have been combined with other deep learning models, such as convolutional neural networks (CNNs) and recurrent neural networks (RNNs), to improve their performance on various tasks. For example, [28] combined VAEs with CNNs to extract hierarchical features from images. It is worth mentioning the characteristic of variational autoencoders is incorporated into our model. In summary, VAEs are powerful for generative modeling and have contributed significantly to the field of deep learning. Deep learning with nonparametric clustering is a pioneer work in applying deep belief network to deep clustering. But in deep clustering based on the probabilistic graphical model, more research comes from the application of variational autoencoder (VAE), which combines variational inference and deep autoencoder together.

**Multi-View Clustering** In practical clustering tasks, the input data usually have multiple views. Multi-view clustering (MVC) methods are proposed to address the limitations of traditional clustering methods by leveraging the complementary information provided by different views of the same instance. Numerous algorithms and techniques have been proposed for MVC, ranging from classic methods such as spectral clustering to more recent deep learning and probabilistic modeling developments. One popular approach is co-training, which was first introduced by [4]. Co-training involves training multiple models, each using a different view of the data, and then using the models to label additional unlabeled data. This approach has been successfully applied in various domains, including text classification [56], image recognition [21], and community detection [24]. Another popular approach is low-rank matrix factorization, which aims to learn a shared latent representation of the data across multiple views. This approach has been explored in various forms, such as structured low-rank matrix factorization [55], tensor-based factorization [41], and deep matrix factorization [54]. Subspace methods have been utilized in MVC in the previous few years, e.g., [7, 13, 26, 30] integrated multiple affinity graphs into a consensus one with the topological relevance considered.

In recent years, deep learning has emerged as a powerful tool for MVC, with various deep neural network architectures proposed for the task. For example, [23] proposed a deep adversarial MVC method, which learns a joint representation of the data across multiple views using adversarial training. [46] proposed deep embedded clustering (DEC) to learn a mapping from the high original feature space to a lower-dimensional one in which a pratical objective is optimized. [48] proposed a framework of multi-level feature learning for contrastive multi-view clustering (MFLVC), which combines MVC with contrastive learning to improve clustering effectiveness.

## 3 The Proposed Method

Self-supervised learning methods for MVC [9, 52, 57, 58] aim at reducing redundant information and achieving sufficient representation from multiple views. In the multi-view scenario, there are also many methods [38, 49] that explore consistent information between views. Despite great efforts, the task of minimizing redundancy and extracting valuable information for subsequent processes while maintaining coherence across perspectives continues to pose significant challenges. In this section, inspired by [9, 39], we extend the principle of the information bottleneck, utilize the reparameterization trick for solving the proposed lower bound and introduce the collaborative training strategy to the MVC scenario. Moreover, we conduct bayes error rates for learned representations to validate the generalization performance of our method.

**Problem Statement** Given the multi-view data, $X$, which consists of $v$ subsets such that $X^i = [x_1^i; \ldots; x_n^i] \in \mathbb{R}^{n \times d_i}$ is the $n \times d_i$-dimensional data of the $i$-th view, where $n$ is the number of instances and $d_i$ is the number of dimensions of the $i$-th view. MVC aims to partition the $n$ instances into $K$ clusters.

### 3.1 Preliminaries

**Analysis of redundancy** According to [1, 9], we can assume that **z** contains all necessary information for predicting **y** without having prior knowledge of **y**. This can be achieved by maintaining all the shared information between the two views $x^i$ and $x^j$. This assumption is grounded on the belief that both views provide similar predictive information. The following formal definition elucidates this concept (See details in Appendix B.2.).

**Definition 3.1** *[9]* $x^i$ *is considered redundant with respect to* $x^j$ *for* **y** *if and only if the mutual information* $I(\mathbf{y}; x^i | x^j) = 0$.

Intuitively, $x^i$ is considered redundant for a task if it is irrelevant for the prediction of **y** when $x^j$ is already observed. Whenever $x^i$ and $x^j$ are mutually redundant with respect to one another ($x^i$ is redundant with respect to $x^j$ for **y**), we have the following:

**Corollary 3.1** *Let* $x^i$ *and* $x^j$ *be two mutually redundant views for a target* **y** *and let* $z^i$ *be a representation of* $x^i$. *If* $z^i$ *is sufficient for* $x^j$ *(*$I(x^i; x^j | z^i) = 0$*) then* $z^i$ *is as predictive for* **y** *as the combination of the two views (*$I(x^i x^j; \mathbf{y}) = I(\mathbf{y}; z^i)$*).*

**Variational bound** The variational bound [18], also known as the Evidence Lower Bound (ELBO), is a widely used optimization objective in Bayesian inference and machine learning. It provides a lower bound on the marginal likelihood of the observed data. Variational bound provides an efficient approximate posterior inference of the latent variable $z^i$ given an observed value $x^i$, which can be formulated as:

$$\log q_{\phi^i}(x^i) \geq L(\phi^i, \theta^i; x^i) = \mathbb{E}_{p_{\theta^i}(z^i|x^i)}[-\log p_{\theta^i}(z^i|x^i) + \log q_{\phi^i}(x^i, z^i)], \tag{1}$$

where $\theta^i$ and $\phi^i$ are generative and variational parameters of the $i$-th view respectively. Through optimizing the lower bound $L(\phi^i, \theta^i; x^i)$, we can approximate the posterior inference of the latent variable $z^i$ accurately based on an observed value of $x^i$. As such, we can obtain more representative embedded features.

### 3.2 SCMVC: Learning Consistent Information via Varational Lower Bound

Variational inference has greatly contributed to representation learning. Introducing variational inference allows representation learning to be performed in a probabilistic manner thereby providing a principled framework for capturing uncertainty in the learned representation. Variational inference can optimize the generative model, resulting in a more informative data representation. In this section, we analyze the consistent information among views from the vartional inference perspective and propose SCMVC (simple consistent multi-view clustering). We introduce a consistent varitional lower bound to model the latent space distribution of input data more consistent with all views. Given $v$-views and global embedded features $\vec{z} = \{z^1, z^2, \ldots, z^v\}$, we formalize this concept as the following theorem (Proofs are given in Appendix B.3.):

**Theorem 3.1 (Consistent Variational Lower Bound)** *Define the generative model* $q_\phi(x^i | \vec{z}, y^i)$, *and a posterior as* $p_\theta(\vec{z}, y^i | x^i) = p_\theta(y^i | \vec{z}) p_\theta(\vec{z} | x^i)$. *The ELBO for multi-view model can be formulated as*

$$\max L_{con}^i = \mathbb{E}_{p_\theta(\vec{z}, y | x^i)}[\log q_\phi(x^i | \vec{z}, y^i)] - \gamma D_{KL}(p_\theta(\vec{z}, y^i | x^i) \| q_\phi(\vec{z}, y^i)), \tag{2}$$

*where the* $y^i$ *is the pseudo-label of* $x^i$, $\vec{z} = \{z^1, z^2 \ldots, z^v\}$, *the first term is to maximize the reconstruction probability of input* $x^i$, *i.e., maximize the decoding probability from latent variable* $\vec{z}$ *to* $x^i$. *The constraint term is the KL divergence of the approximate from the true posterior to ensure that the distribution of acquired features and labels is consistent with their intractable true posterior. It can be intuitively understood that our goal is to obtain the unique distribution of the embedded feature* $\vec{z}$ *of x for different categories.*

For better understanding, Eq. (2) can be expanded out as:

$$\max L_{con}^i = \mathbb{E}_{x^i \sim \tilde{p}(x^i)}[\mathbb{E}_{\vec{z} \sim \tilde{p}(\vec{z}|x^i)}[\log q_\phi(x^i|\vec{z}) - \gamma(\sum_{i=1}^{v} \sum_{y} p_\theta(y^i|\vec{z}) \log \frac{p_\theta(\vec{z}|x^i)}{q_\phi(\vec{z}|y^i)} + KL(p_\theta(y^i|\vec{z}) \| q_\phi(y^i)))]], \tag{3}$$

Overall, the objective function of SCMVC is:

$$\max \sum_{i=1}^{v}(L_{con}^i) = \sum_{i=1}^{v}(L_{rec}^i - \gamma L_{KL}^i), \tag{4}$$

where $L_{rec}^i$ is the reconstruction term and $L_{KL}^i$ is the constraint term which approximates the true posterior.

### 3.3 SUMVC: Learning Sufficient Representation via Reducing Redundancy

Although SCMVC ensures consistency among views, there is a lack of consideration for reducing mutual redundancy and achieving sufficiency among views, which is important to obtain latent representations suitable for downstream tasks. Next, we advance our work and extend the information bottleneck principle to the multi-view setting. By decomposing the mutual information between $x^i$ and $z^i$ (See Appendix B.1.), we can identify two components:

$$I_{\theta_i}(x^i, z^i) = I_{\theta_i}(z^i, x^i|x^j) + I_{\theta_i}(x^j, z^i),$$

where the first term denotes superfluous information and the second term denotes signal-relevant information. We take view $j$ as self-supervised signals to guide the learning of view $i$. To ensure that the representation is sufficient for the task, we maximize the mutual information between $x^j$ and $z^i$. As for $I_{\theta_i}(z^i, x^i|x^j)$, this term indicates the information in $z^i$ that is exclusive to $x^i$ and cannot be predicted by observing $x^j$. As we assume mutual redundancy between the two views, this information is deemed irrelevant to the task and could be safely discarded [9]. Overall, to extract sufficient representation of $z^i$, we maximize the following formulation:

$$\max -I_{\theta_i}(z^i, x^i|x^j) + I_{\theta_i}(x^j, z^i), \tag{5}$$

which can be optimized via maiximazing the lower bound in Theorem 3.2.

**Theorem 3.2 (Sufficient Representation Lower Bound)** *Considering a multi-view model, the generative model is $q_{\phi^i}(x^i|z^i, y^i)$. The sufficient representation lower bound can be defined as:*

$$\max L_{suf}^{ij} = -D_{KL}(p_{\theta^i}(z^i|x^i), p_{\theta^j}(z^j|x^j)) + I_{\theta^i\theta^j}(z^i, z^j). \tag{6}$$

**Proof 1** *Considering $z^i$ and $z^j$ on the same domain $\mathbb{Z}$, we have*

$$-I_\theta(x^i; z^i|x^j) = -\mathbb{E}_{x^i, x^j \sim p(x^i, x^j)}\mathbb{E}_{z \sim p_{\theta^i}(z^i|x^i)}\left[\log \frac{p_{\theta^i}(z^i = z|x^i = x^i)}{p_{\theta^i}(z^i = z|x^j = x^j)}\right] \tag{7}$$

$$= -\mathbb{E}_{x^i, x^j \sim p(x^i, x^j)}\mathbb{E}_{z \sim p_{\theta^i}(z^i|x^i)}\left[\log \frac{p_{\theta^i}(z^i = z|x^i = x^i)p_{\theta^j}(z^j = z|x^j = x^j)}{p_{\theta^j}(z^j = z|x^j = x^j)p_{\theta^i}(z^i = z|x^j = x^j)}\right] \tag{8}$$

$$= -\mathbb{E}_{x^i, x^j \sim p(x^i, x^j)}\left[D_{KL}(p_{\theta^i}(z^i|x^i)\|p_{\theta^j}(z^j|x^j)) - D_{KL}(p_{\theta^i}(z^i|x^j)\|p_{\theta^j}(z^j|x^j))\right] \tag{9}$$

$$\geq -D_{KL}(p_{\theta^i}(z^i|x^i)\|p_{\theta^j}(z^j|x^j)). \tag{10}$$

*For $I_{\theta^i}(x^j, z^i)$, it can be lower bounded as:*

$$I_{\theta^i}(x^j, z^i) \stackrel{(P2)}{=} I_{\theta^i\theta^j}(z^i; z^j x^j) - I_{\theta^i\theta^j}(z^i; z^j|x^j) \; = I_{\theta^i\theta^j}(z^i; z^j x^j) \; = I_{\theta^i\theta^j}(z^i; z^j) + I_{\theta^i\theta^j}(z^i; x^j|z^j)$$
$$\geq I_{\theta^i\theta^j}(z^i; z^j),$$

*where $I_{\theta^i\theta^j}(z^i; x^j|z^j)$=0 when $z^j$ is sufficient for $z^i$. This occurs whenever $z^j$ contains all the necessary information about $z^i$.*

*Therefore, Eq. (5) can be optimized via maximizing the lower bound:*

$$-D_{KL}(p_{\theta^i}(z^i|x^i), p_{\theta^j}(z^j|x^j)) + I_{\theta^i\theta^j}(z^i, z^j).$$

In addition, we adopt collaborative training to allow mutual supervision to better adapt to multi-view scenes. The views are selected in turn to act as the supervisory view, while the remaining views serve as the supervised views. This learning approach could efficiently use the common information and complementary information of multiple views.

SUMVC is derived by incorporating a lower bound on sufficient representation into SCMVC, the objective of which can be defined as follows:

$$\max \sum_{i=1}^{v} (L_{con}^i + \sum_{j=1,j\neq i}^{v} \beta L_{suf}^{ij}), \tag{11}$$

where the $i$-th view can be regarded as the supervisory view, while the $j$-th view is the supervised view. Through the collaborative training we could reduce redundancy and obtain consistent information among views. What's more, we align the embedded features and category distribution with the original distribution to obtain information conducive to clustering. Finally, in order to better utilize global features, we run $K$-means on $\vec{z}$ to obtain the clustering results.

### 3.4 The Reparameterization Trick for Problem Solving

In order to solve the problem, let $z^i$ be a continuous random variable, and $z^i \sim p_{\theta^i}(z^i|x^i)$ be some conditional distribution of the $i$-th view. It is then often possible to express the random variable $z^i$ as a deterministic variable $z^i = f_{\theta^i}(\epsilon^i, x^i)$, where $\epsilon^i$ is an auxiliary variable with independent marginal $q(\epsilon)$ and function $f_{\theta^i}(\cdot)$ parameterized by $\theta^i$. We let $z^i \sim q_{\phi^i}(z^i|x^i) = N(\mu^i, \sigma^{i2})$. In this case, a valid reparameterization is $z^i = \mu^i + \sigma^i \epsilon^i$, where $\epsilon^i$ is an auxiliary noise variable $\epsilon^i \sim N(0, 1)$.

## 4 Discussion

### 4.1 Connection to VaDE

Variational deep embedding (VaDE) [17] is an unsupervised generative clustering algorithm, which combines VAE and Gaussian Mixture Model (GMM) to model the data generation process. In our method, from Eq. (3), it can be found that SCMVC has taken into account the consistency information among views compared with the VaDE for the single-view scenario. Moreover, we introduce the SUMVC to help obtain a more sufficient representation, and we propose new methods to solve the problem based on the reparameterization trick.

### 4.2 Bayes Error Rate for Downstream Task

In this section, inspired by [8, 39], we present a theoretical analysis of the generalization error of representations, assuming $T$ to be a categorical variable. We take view $j$ as self-supervised signals to guide the learning of view $i$ as an example. To represent the irreducible error when inferring labels from the representation via an arbitrary classifier, we use Bayes error rate $P_e := \mathbb{E}_{z^i \sim P_{z^i}}[1 - \max_{t \in T} P(\hat{T} = t|z^i)]$, where $P_e$ is the Bayes error rate of arbitrary learned representations $z^i$, and $\hat{T}$ is the estimated label $T$ from our classifier. Analyzing the Bayes error rate allows us to gain insights into the generalization performance of our propose model. We present a general form of the complexity of the sample with mutual information $I(z^i; x^j)$ estimation using empirical samples from the distribution $P_{z^i,x^j}$. Let $P_{z^i,x^j}^{(n)}$ denote the uniformly sampled empirical distribution of $P_{z^i,x^j}$. We define $\hat{I}_{\theta^*}^{(n)}(z^i; x^j)$ as $\mathbb{E}_{P_{z^i,x^j}^{(n)}}[\log \frac{p(x^j|z^i)}{p(x^j)}]$.

**Proposition 1 (Mutual Information Neural Estimation [40].)** *Let $0 < \delta < 1$. There exists $d \in \mathbb{N}$ and a family of neural networks $F := \hat{f}_\theta : \theta \in \Theta \subseteq \mathbb{R}^d$, where $\Theta$ is compact, so that $\exists \theta^* \in \Theta$, with probability at least $1 - \delta$ over the draw of $\{z_{x_m^i}, x_m^j\}_{m=1}^n \sim P_{z^i,x^j}^{\otimes n}$,*

$$|\hat{I}_{\theta^*}^{(n)}(z^i; x^j) - I(z^i; x^j)| \leq O(\sqrt{\frac{1 + log(\frac{1}{\delta})}{n}}). \tag{12}$$

This proposition shows that there exists a neural network $\theta^*$, with high probability, $\hat{I}_{\theta^*}^{(n)}(z^i; x^j)$ can approximate $I(z^i; x^j)$ with $n$ samples at rate $O\left(\frac{1}{\sqrt{n}}\right)$. Under this network $\theta^*$ and the same parameters $d$ and $\delta$, we are ready to present our main results on the Bayes error rate.

**Theorem 4.1 (Bayes Error Rates for Arbitrary Learned Representations [39].)** *Let $|T|$ be $T$'s cardinalitiy and $Th(x) = \min\{\max\{x, 0\}, 1 - \frac{1}{|T|}\}$. Given $p_e = Th(\bar{p}_e)$, we have a thresholding function:*

$$\bar{p}_e \leq 1 - exp(-(H(T) + I(x^i; x^j|T) + I(z^i; x^i|x^j, T) - \hat{I}_{\theta^*}^{(n)}(z^i; x^j) + O(\sqrt{\frac{1 + log(\frac{1}{\delta})}{n}})). \quad (13)$$

Given arbitrary learned representations ($z^i$), Theorem 4.1 suggests the corresponding Bayes error rate ($P_e$) is small when: 1) the estimated mutual information $\hat{I}_{\theta^*}^{(n)}(z^i; x^j)$ is large; 2) a larger number of samples are used for estimating the mutual information; and 3) the task-irrelevant information $I(z^i; x^i|x^j, T)$ is small. The first and the second results support the claim that maximizing $I(z^i; x^j)$ may learn the representations beneficial to downstream tasks. The third result implies the learned representations may perform better on the downstream task when the task-irrelevant information is small. We evaluate the quality of our self-supervised method's learned representations using Bayes Error Rates in Theorem 4.2. The proofs are given in Appendix B.4.

**Theorem 4.2** *The Bayes Error Rates for arbitrary learned representations can be minimized by maximizing $L_1^i + \beta L_2^{ij}$, formally:*

$$\max L_{con}^i + \beta L_{suf}^{ij} \leftrightarrow \min \bar{p}_e. \quad (14)$$

## 5 Experiments

### 5.1 Experimental Setup

Table 1: The statistics of experimental datasets.

| Dataset | #Samples | #Views | #Clusters |
|---|---|---|---|
| Multi-MNIST | 70000 | 2 | 10 |
| Multi-Fashion | 10000 | 3 | 10 |
| Multi-COIL-10 | 720 | 3 | 10 |
| Multi-COIL-20 | 1440 | 3 | 20 |

**Datasets** As shown in Table 1, we use the following four real-world multi-view datasets in our study. MNIST [20] is a widely used dataset of handwritten digits from 0 to 9. The Fashion dataset [45] comprises images of various fashion items, including T-shirts, dresses, coats, etc. The COIL dataset [29] contains images of various objects, such as cups, ducks, and blocks, shown in different poses. We use multi-view datasets derived from origin datasets: Multi-COIL-10, Multi-COIL-20, Multi-MNIST and Multi-Fashion. Each dataset includes multiple views of each example, all randomly sampled from the same category. In Multi-COIL-10 (K = 10) and Multi-COIL-20 (K = 20), different views of an object correspond to various poses, but retain the same label. In Multi-MNIST, different views of a digit represent the same digit written by different individuals. In Multi-Fashion, different views of a product category signify different fashionable designs for that category.

**Comparison Methods** The comparison methods include three single-view clustering methods: $K$-means (1967) [27], $\beta$ -VAE ($\beta$-VAE: learning basic visual concepts with a constrained variational framework (2017) [12]), and VaDE (variational deep embedding: an unsupervised and generative approach to clustering (2016) [17]), the input of which is the concatenation of all views, and five state-of-the-art MVC methods: BMVC (binary multi-view clustering (2018) [53]), SAMVC (self-paced and auto-weighted multi-view clustering (2020) [32]), RMSL (reciprocal multi-layer subspace learning for multi-view clustering (2019) [22]), DEMVC (deep embedded multi-view clustering with collaborative training (2021) [47]), FMVACC (fast multi-view anchor-correspondence clustering (2022) [43]).

**Evaluation Measures** Three measures are used to evaluate the clustering results, including, clustering accuracy (ACC), normalized mutual information (NMI), and adjusted rand index (ARI). A larger value of these metrics indicates a better clustering performance.

Table 2: Clustering results of all methods on four datasets. The best result in each row is shown in bold and the second-best is underlined.

| Datasets | $K$-means | $\beta$-VAE | VaDE | BMVC | SAMVC | RMSL | DEMVC | FMVACC | **SCMVC** | **SUMVC** |
|---|---|---|---|---|---|---|---|---|---|---|
| | | | | | ACC | | | | | |
| Multi-MNIST | 53.9 | 49.3 | 30.7 | 89.3 | - | - | 98.2 | 55.6 | 94.5 | **99.8** |
| Multi-Fashion | 47.6 | 51.3 | 40.6 | 62.2 | 77.9 | 77.9 | 78.6 | 77.4 | 84.9 | **91.4** |
| Multi-COIL-10 | 73.3 | 59.8 | 32.5 | 66.7 | 67.8 | 96.4 | 89.1 | 93.2 | 98.1 | **100.0** |
| Multi-COIL-20 | 41.5 | 53.1 | 20.3 | 57.0 | 83.4 | 66.5 | 85.0 | 75.8 | 83.7 | **100.0** |
| | | | | | NMI | | | | | |
| Multi-MNIST | 48.2 | 43.6 | 35.4 | 90.2 | - | - | 98.9 | 48.2 | 87.3 | **99.3** |
| Multi-Fashion | 51.3 | 51.0 | 53.7 | 75.6 | 68.8 | 75.6 | **90.3** | 73.8 | 80.9 | 87.5 |
| Multi-COIL-10 | 76.9 | 68.5 | 44.8 | 68.1 | 82.6 | 92.5 | 89.7 | 93.4 | 96.7 | **100.0** |
| Multi-COIL-20 | 64.5 | 66.7 | 36.9 | 90.0 | 79.1 | 76.3 | 96.5 | 84.9 | 90.8 | **100.0** |
| | | | | | ARI | | | | | |
| Multi-MNIST | 36.0 | 29.1 | 8.5 | 85.6 | - | - | 98.6 | 45.2 | 88.5 | **99.5** |
| Multi-Fashion | 34.8 | 33.7 | 22.8 | 68.2 | 55.7 | 68.2 | 77.2 | 70.3 | 75.6 | **84.1** |
| Multi-COIL-10 | 64.8 | 51.4 | 18.1 | 53.0 | 62.1 | 92.1 | 89.7 | 92.5 | 95.8 | **100.0** |
| Multi-COIL-20 | 38.4 | 45.0 | 9.0 | 81.3 | 55.4 | 58.7 | 86.0 | 79.1 | 79.9 | **100.0** |

Table 3: Ablation studies of SUMVC on loss components.

| Components | | | Multi-MNIST | | | Multi-Fasion | | | Multi-COIL-20 | | |
|---|---|---|---|---|---|---|---|---|---|---|---|
| $L_{rec}$ | $L_{KL}$ | $L_{suf}$ | ACC | NMI | ARI | ACC | NMI | ARI | ACC | NMI | ARI |
| ✓ | ✗ | ✗ | 70.5 | 72.9 | 62.6 | 56.7 | 57.6 | 47.7 | 62.5 | 81.1 | 63.3 |
| ✓ | ✓ | ✗ | 94.5 | 87.3 | 88.5 | 84.9 | 80.9 | 75.6 | 83.7 | 90.8 | 79.9 |
| ✗ | ✗ | ✓ | 11.5 | 0.2 | 0.1 | 11.3 | 0.1 | 0.1 | 9.9 | 4.6 | 0.0 |
| ✓ | ✓ | ✓ | 99.8 | 99.3 | 99.5 | 91.4 | 87.5 | 84.1 | 100.0 | 100.0 | 100.0 |

## 5.2 Comparison with State-of-the-Art Methods

Table 2 shows the quantitative comparison between the proposed methods and baseline models for several datasets. The best result is highlighted in bold, while the second-best result is underlined in each line. Despite its simplicity, the SCMVC method demonstrates similar performance to existing methods in most cases. This is primarily because our method focuses on extracting global information from multiple distributions. Moreover, the results reveal that SUMVC outperforms existing methods in terms of three evaluation methods. Notably, SUMVC achieves remarkable improvements on Multi-COIL-10 and Multi-COIL-20. The reason is that the proposed lower bound could explore consistent predictive information and reduce redundancy among views, which benefits clustering results.

## 5.3 Ablation Studies

To further verify the contributions of the proposed method, we conduct ablation studies on Eq. (11), which consists of $L_{rec}$, $L_{KL}$, and $L_{suf}$. As shown in Table 3, it is found that the performance of clustering is significantly influenced by $L_{KL}$. We can also find that combining $L_{con}$ and $L_{suf}$ yields better results than using either $L_{con}$ or $L_{suf}$ alone, which illustrates that each loss function plays a crucial role in the overall performance of the model. We find that the clustering performance is not ideal when only $L_{suf}$ is used for training. The reason is that the latent features extracted by the lack of reconstruction constraints cannot accurately reflect the information of the original features.

## 5.4 Visualization of Feature Representation

We use $t$-SNE [42] to visualize the learning process of embedded features. The non-linear dimensionality reduction technique is used for visualizing and clustering high-dimensional data, which has been widely used in machine learning and data analysis to visualize high-dimensional data

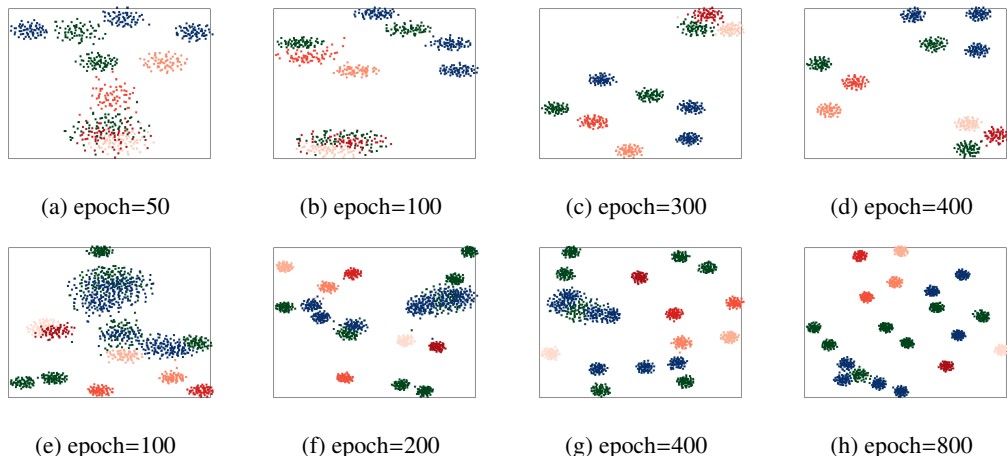

| (a) epoch=50 | (b) epoch=100 | (c) epoch=300 | (d) epoch=400 |

| (e) epoch=100 | (f) epoch=200 | (g) epoch=400 | (h) epoch=800 |

Figure 1: Visualization of the latent representations $\vec{z}$ in training on Multi-COIL-10 and Multi-COIL-20 in the first and second rows, respectively.

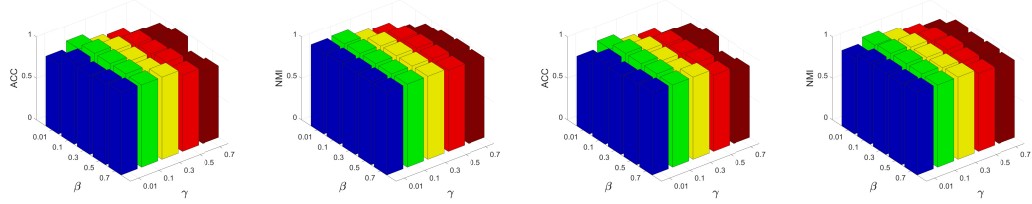

Figure 2: Clustering performance w.r.t. different parameter settings on Multi-COIL-10 and Multi-COIL-20 respectively.

in lower-dimensional spaces for easier understanding and interpretation. We show the embedded features $\vec{z}$ of Multi-COIL-10 and Multi-COIL-20 in Fig. 1. At the start of the training stage, the embedded features are non-separable. As the training progresses, the clustering structures of the embedded features become more apparent while their centroids gradually separate. This visual demonstration highlights the effectiveness of our proposed model.

### 5.5 Parameter Sensitivity Analysis

SUMVC has two primary hyperparameters, i.e., $\gamma$ and $\beta$. We test the clustering performance with different settings to demonstrate the stability of SUMVC. Results are presented in Fig. 2. Additionally, higher values of $\gamma$ are required to encourage disentangling but result in a fidelity-disentanglement trade-off between the quality of reconstructions and its latent features disentanglement. $\beta$ that is either too large or too small also leads to reduced clustering performance due to the imbalance between reducing redundant information and obtaining consistent information. The recommended parameters for $\gamma$ and $\beta$ are both 0.1 in our experiments.

## 6 Conclusion

In this paper, we propose a simple consistent approach and an information-theoretic approach for multi-view clustering to address the challenge of obtaining useful information from multiple views while reducing redundancy and ensuring consistency among views. In particular, we leverage the information bottleneck principle and variational inference to the multi-view scenarios. The proposed SCMVC and SUMVC are extensively evaluated on multiple real-world multi-view datasets. Extensive experiments demonstrate the clustering performance of the proposed models.

## Acknowledgement

This work is supported in part by Medico-Engineering Cooperation Funds from University of Electronic Science and Technology of China (ZYGX2021YGLH022). Lifang He is partially supported by the National Science Foundation grants (MRI-2215789, IIS-1909879, IIS-2319451) and Lehigh's grants under Accelerator and CORE.

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
