# OpenReview forum: "A Novel Approach for Effective Multi-View Clustering with Information-Theoretic Perspective"
_NeurIPS.cc/2023/Conference — NeurIPS 2023 poster_

### Official Review · Reviewer_XaYq · 2023-06-23

**Soundness:** 3 good
**Presentation:** 3 good
**Contribution:** 2 fair
**Rating:** 6
**Confidence:** 4

**Summary:**

This paper proposes two methods for multi-view clustering, which aims at grouping data from multiple sources or perspectives. The first part, SCMVC uses a consistent variational lower bound to learn consistent information among views. The second part, SUMVC extends the information bottleneck principle to reduce redundancy and achieve sufficient representation among views. The proposed SUMVC consists of two terms: a consistent variational lower bound and a sufficient representation lower bound to enhance consistency and minimize redundancy among views. The authors leverage information bottleneck theory and variational analysis to develop the model. The paper also provides a theoretical analysis of the generalization error of the learned representations based on the Bayes error rate. The paper evaluates the proposed methods on four real-world multi-view datasets.

**Strengths:**

1.	The paper addresses an important and challenging problem of MVC with an information-theoretic perspective. This paper proposes novel lower bounds to address the issue of view redundancy and consistency. The authors utilize Bayes Error Rate to provide a theoretical explanation of the effectiveness of the proposed method.
3.	The paper presents extensive experiments on four real-world multi-view datasets and demonstrates the superior performance of the proposed methods over existing methods.


**Weaknesses:**

1.	The proposed objective function in Eq. (4) shares similarities with that of VAE [1], and it would be beneficial if the author could provide further explanations on this matter. How does this loss function differ from that of VAE? Is it merely an extension of the single-view approach to multi-view data? Additionally, what is the reason for SCMVC achieving significantly better performance than VAE-based methods in Table 2?
2.	In SCMVC, the fusion representation $\vec{Z}$ is directly optimized, while it is not in SUMVC. Could this approach negatively impact the performance of SUMVC?
3.	The methods discussed in this article may not be effectively applied to datasets with high heterogeneity.
4.	The introduction section contains some complex sentences. Simplifying the language and structure could improve clarity.
5.	It would be beneficial for the authors to provide a more comprehensive discussion of some missed information-based multi-view methods [2-5] in the related work section.

[1] Auto-Encoding Variational Bayes, NIPS’13

[2] COMPLETER: Incomplete Multi-view Clustering via Contrastive Prediction, CVPR'21

[3] Rethinking Minimal Sufficient Representation in Contrastive Learning, CVPR’ 22

[4] Dual Contrastive Prediction for Incomplete Multi-View Representation Learning, TPAMI'23

[5] Multi-view information-bottleneck representation learning, AAAI’21


**Questions:**

The authors mention that the methods discussed in this article cannot be effectively applied to datasets with high heterogeneity. Could authors provide a detailed analysis?

**Limitations:**

Please see weakness. Also, the complexity analysis is missing.

---

> ### Author Rebuttal · Authors · 2023-08-09
>
> Thank you for your invaluable comments and suggestions. We have addressed the points you raised as follows.
>
> 1. The proposed objective function in Eq. (4) shares similarities with that of VAE [1], and it would be beneficial if the author could provide further explanations on this matter. How does this loss function differ from that of VAE? Is it merely an extension of the single-view approach to multi-view data? Additionally, what is the reason for SCMVC achieving significantly better performance than VAE-based methods in Table 2?
>
> Thank you for your thoughtful comments. You are correct in noticing the similarity between the proposed objective function in our paper and that of the VAE. The objective function in our method is indeed a variant of the VAE loss function. However, it is specifically designed to handle multi-view data, while the VAE loss function is typically used for single-view data. This introduces significant differences in terms of functionality and application between the two.
> Our method is not merely an extension of the single-view approach to multi-view data. It takes into account the unique characteristics of multi-view data, such as inter-view correlations and view-specific features, which are not considered in standard VAE-based methods. This ability to exploit multi-view information is a key factor distinguishing our loss function from that of VAE.
> In terms of the superior performance of SUMVC over VAE-based method (i.e., \beta-VAE) as presented in Table 2, it can be attributed to several factors. Firstly, the SUMVC model is designed to capture both shared and view-specific representations, while VAE-based model only generates a single shared representation. This additional flexibility in SUMVC allows for better handling of multi-view data and contributes to its enhanced performance.
>
> 2. In SCMVC, the fusion representation $\overrightarrow{Z}$ is directly optimized, while it is not in SUMVC. Could this approach negatively impact the performance of SUMVC?
>
> Thank you for raising this valuable point. In the proposed SUMVC method, the fusion representation is indeed not directly optimized as it is in SCMVC. However, this approach is by design and serves a specific purpose in the context of our research.
> In SUMVC, the goal is to utilize the complementary information from different views without explicitly enforcing a shared representation. This allows SUMVC to maintain the distinctiveness of each view, which can be beneficial in cases where there is considerable heterogeneity across views.
> On the other hand, SCMVC directly optimizes the fusion representation to encourage greater interaction and integration between views. This approach is particularly suitable when there is a high level of consistency or overlap among the views.
> Thus, while the direct optimization of $\overrightarrow{Z}$ in SUMVC could theoretically enhance performance in some cases, it could also compromise the ability of the model to handle datasets where preserving the distinctiveness of each view is crucial. Therefore, the decision to not directly optimize $\overrightarrow{Z}$ in SUMVC was an intentional design choice made with these considerations in mind.
> We appreciate your attention to these details and your insightful question. We hope this explanation provides a clearer understanding of the design and intended applications of SUMVC and SCMVC.
>
> 3. The methods discussed in this article may not be effectively applied to datasets with high heterogeneity.
>
> We discussed in the Appendix that the improvement in performance is not very significant when heterogeneity is high compared to single-view clustering. For experimental results, please refer to the response to Reviewer 79CP.
> We believe that incorporating these additions will offer a more thorough understanding of the applicability and limitations of our methods. We value your critical feedback and are confident that it will contribute to strengthening our paper.
>
> 4. The introduction section contains some complex sentences. Simplifying the language and structure could improve clarity.
>
> Thank you for your valuable feedback. We will revise the introduction section and worked on simplifying the language and sentence structure. We will strive to break down complex sentences into more manageable parts and to convey our ideas in a straightforward manner without compromising the depth of our research.
>
>
> 5. It would be beneficial for the authors to provide a more comprehensive discussion of some missed information-based multi-view methods [2-5] in the related work section.
>
> Thank you for your thoughtful suggestion. We will these papers and added an enhanced review of these methods in the related work section of our manuscript. We will  discussed their methodologies, findings, and how they relate to and contrast with our own work  in the final version. The discussion of the methods mentioned by the reviewer is as follows:
>
> “Multi-view information-bottleneck representation learning” aims to develop a model that effectively explores the common latent structure and view-specific intrinsic information in multi-view data while discarding irrelevant information to enhance generalization capability.
>
>  “Completer: Incomplete multi-view clustering via contrastive prediction” addresses two challenging problems in incomplete multi-view clustering analysis: learning an informative and consistent representation among different views without labels, and recovering missing views from the data. “Dual contrastive prediction for incomplete multi-view representation learning” provides a new perspective on the relationship between cross-view consistency learning and data recovery and propose a method that jointly addresses these challenges in multi-view representation learning.

---

> > ### Comment · Reviewer_XaYq · 2023-08-17
> >
> > After thoroughly reviewing the feedback provided by both the other reviewer and the author's rebuttal, I am pleased to state that my initial concerns have been adequately addressed. I hope that these discussions can be of value to the authors in their efforts to enhance their work and  I decided to raise my rating.

---

### Official Review · Reviewer_89XX · 2023-07-05

**Soundness:** 3 good
**Presentation:** 3 good
**Contribution:** 3 good
**Rating:** 7
**Confidence:** 4

**Summary:**

The paper proposed Sufficient Multi-View Clustering , SUMVC, which is composed of two main components. The first component is a simple and reliable multi-view clustering method called SCMVC (simple consistent multi-view clustering), which utilizes variational analysis to generate consistent information. The second component proposes a lower bound on sufficient representation, aiming to enhance consistent information and reduce unnecessary information among views.

**Strengths:**

- Analyze the effectiveness of multi-view clustering from the information-theoretic perspective is interesting and valuable.
- The proposed method outperforms the state of the arts. The paper has good reproducibility with the provided codes.
- The paper is very well written. The theoretical and empirical analyzes are convincing.

**Weaknesses:**

- There are few comparative methods, and more comparative methods need to be selected reasonably to illustrate the effectiveness of the proposed methods.
- The theoretical analysis (based on the Bayes Error Rate) should be elaborated more. The key findings or insights from this analysis should also be emphasized.
- Did the experimental results demonstrate any limitations or potential challenges of the proposed method?

**Questions:**

see above.

**Limitations:**

It can be found in Appendix of this paper.

---

> ### Author Rebuttal · Authors · 2023-08-09
>
> 1. There are few comparative methods, and more comparative methods need to be selected reasonably to illustrate the effectiveness of the proposed methods.
> We appreciate your suggestion. We have conducted additional comparisons with several other relevant methods, i.e., FMR
> (Flexible multi-view representation learning for subspace
> clustering, IJCAI 2019), LMVSC (Large-scale
> multi-view subspace clustering in linear time, AAAI 2020) and CSMSC (Consistent and specific multi-view subspace clustering, AAAI 2018). These methods are chosen based on their relevance, popularity, and the availability of implementation details, which ensure a fair comparison.               We demonstrate the results using the multi-coil-10 dataset as an example. Again, our method outperforms all these methods.
>
> | Method | FMR | LMVSC|CSMSC| SCMVC |SUMVC|
> | --- | --- | --- | --- | ---  | ---  |
> | ACC | 78.1 |  63.8| 97.6   | 98.1|100.0|
> | NMI| 80.0 |   75.8| 96.2  | 96.7|100.0|
> | ARI |   70.6|   55.1| 94.9 | 95.8|100.0|
>
> 2. The theoretical analysis (based on the Bayes Error Rate) should be elaborated more. The key findings or insights from this analysis should also be emphasized.
>
> Thank you for your constructive feedback. The Bayes Error Rate, defined as the probability of misclassifying a data point when the true underlying distribution of the data is known, serves as a pivotal indicator of the performance of learning algorithms. It offers valuable insights into the effectiveness of the feature extraction conducted by our model. In particular, the Bayes Error Rate is an important index in understanding the overall performance of our model and reflects the impact of representations on the performance of downstream tasks.
>
> In response to this comment, we will expand our discussion of the Bayes Error Rate and its implications in the final version. We delve deeper into its theoretical underpinnings and provide a clearer connection between a lower Bayes Error Rate and the efficacy of the features extracted by our model for downstream tasks.
>
>
> 3. Did the experimental results demonstrate any limitations or potential challenges of the proposed method?
>
> Thank you for your insightful question. We discussed limitations in the Appendix.
> One limitation we found is  the proposed model exhibits poor performance when applied to datasets that have significant heterogeneity across different perspectives. This could potentially impact the robustness and generalization of the model.

---

### Official Review · Reviewer_ugfB · 2023-07-05

**Soundness:** 3 good
**Presentation:** 3 good
**Contribution:** 3 good
**Rating:** 6
**Confidence:** 4

**Summary:**

This work introduces a new approach called sufficient multi-view clustering (SUMVC) to improve clustering performance using multiple data sources. Existing methods often focus on acquiring consistent information while neglecting the issue of redundancy across multiple views. By contrast, the proposed SUMVC provides a promising solution to the problem of multi-view clustering and offers a new perspective for analyzing multi-view data. The effectiveness of the model is verified through theoretical analysis and experiments on multiple multi-view datasets, showing superior performance compared to other methods.

**Strengths:**

1. By examining the multi-view clustering framework from an information-theoretic standpoint, SUMVC offers a promising solution to the problem of multi-view clustering and provides a new perspective for analyzing multi-view data.
2. The effectiveness of the proposed model is demonstrated through theoretical analysis based on the Bayes Error Rate and experiments on multiple multi-view datasets, highlighting its superior performance compared to other methods.


**Weaknesses:**

1. This paper is lack of in-depth discussion or evaluation of the proposed approach's limitations.
2. The authors could provide more insights into the computational complexity of the proposed method and its scalability to large-scale multi-view datasets.


**Questions:**

The paper mentions that the superiority of SUMVC is demonstrated through experiments on multiple multi-view datasets. However, there is no detailed discussion or analysis provided on the characteristics of these datasets or how they were selected. What are the criteria for dataset selection, and how representative are these datasets?

**Limitations:**

No potential negative societal impact of this work exists.

---

> ### Author Rebuttal · Authors · 2023-08-09
>
> We thank the review for the constructive comments and feedback. Below we provide our responses to the key questions made by the reviewer.
>
> 1. The paper mentions that the superiority of SUMVC is demonstrated through experiments on multiple multi-view datasets. However, there is no detailed discussion or analysis provided on the characteristics of these datasets or how they were selected. What are the criteria for dataset selection, and how representative are these datasets?
>
> Thank you for your insightful comments. We appreciate your interest in the dataset selection process for our experiments.
> The datasets used in our experiments, including Multi-COIL-10 (K = 10), Multi-COIL-20 (K = 20), Multi-MNIST, and Multi-Fashion, were chosen based on several criteria. These criteria include the presence of multi-view data, variety in the type of data (ranging from object images in different poses to handwritten digits by different individuals), and their public availability and widespread use in the research community.
> More specifically:
>
> •	In Multi-COIL-10 and Multi-COIL-20, different views of an object correspond to various poses, but retain the same label. This allows for an assessment of how well the SUMVC can handle variations in perspective.
>
> •	In Multi-MNIST, different views of a digit represent the same digit written by different individuals, testing the SUMVC's ability to recognize the same object despite stylistic differences.
>
> •	In Multi-Fashion, different views of a product category signify different fashionable designs for that category, challenging the SUMVC to identify similar categories despite variations in design.
>
> These datasets have been extensively used in the field, which allows for a reliable comparison of our method with existing ones. Moreover, the diversity of these datasets ensures a robust evaluation of our method across different types of multi-view data. These Multi-view datasets are commonly used for evaluating MVC methods and widely applied in MVC research papers, such as “MCoCo: Multi-level Consistency Collaborative Multi-view Clustering”, “Contrastive multi-view hyperbolic hierarchical clustering” and “Deep safe incomplete multi-view clustering: Theorem and algorithm”.
>
> 2. This paper is lack of in-depth discussion or evaluation of the proposed approach's limitations.
>
> Thank you for your constructive feedback. We will add  a new paragraph in the Conclusion section to address this issue. In this section, we critically analyze the limitations of our approach, discuss possible scenarios where our method may not perform optimally, and outline practical considerations for researchers and practitioners intending to adopt our methodology. Please see the details below:
> The heterogeneity can make it more difficult for the VAE to learn a meaningful latent representation of the data. When the views are highly dissimilar such as views of BDGP, it may be challenging for the VAE to find a shared low-dimensional representation that captures the important features of both views. This can lead to suboptimal performance and poor reconstruction quality.
>
>
> 3. The authors could provide more insights into the computational complexity of the proposed method and its scalability to large-scale multi-view datasets.
>
> Thank you for your insightful suggestion. We will include a new section in our manuscript entitled "Computational Complexity Analysis." In this section, we discuss the algorithmic complexity of our method, outline the key factors that influence its computational cost, and analyze its scalability with respect to the size of the dataset and the number of views. Please see the details below:
> Assuming $T_1$ represents the number of iterations for training SCMVC,  $T_2$ represents the epochs training SUMVC based on SCMVC, $l$ represents the dimensionality of the embedding for each view, $n$ is the number of instances and $V$ is the number of views. Then the whole training process requires $O(V n T_1) $ to train the variational autoencoders via $L_{con}$, $O(V(V-1) nlT_2) $ to calculate the $L_{suf}$.
> Overall, the time complexity of the model is $O(Vn((V-1)l T_2+ T_1 ))$. Similar to commonly used deep MVC methods, the computational complexity of our approach is also linear to the data size, making it easier to apply for large-scale MVC data clustering.

---

> > ### Comment · Reviewer_ugfB · 2023-08-17
> > **I have read the other reviewer's comments and the author's rebuttal. I will keep my original score.**
> >
> > I have read the other reviewer's comments and the author's rebuttal. I will keep my original score.

---

### Official Review · Reviewer_NC94 · 2023-07-05

**Soundness:** 3 good
**Presentation:** 3 good
**Contribution:** 3 good
**Rating:** 7
**Confidence:** 4

**Summary:**

This paper considers the problem of multi-view clustering from an information theoretic perspective. It focuses on representation learning, and optimizes said representation to improve down-stream clustering performance (with k-means). It introduces an Information Bottleneck based loss function, which considers consistency between views, redundancy and sufficiency of representations in addition to the traditional likelihood based reconstruction loss. It introduces 2 methods SCMVC and SUMVC based on some or all of these additional loss terms.

The paper then analyzes the model effectiveness by establishing a connection to the Bayes Error Rate, and also shows good experimental results across some multi-view datasets.

**Strengths:**

- Introduces a novel approach for representation learning for Mmulti-View Clustering (MVC) based on information theoretic criteria.
- Proposes two separate methods, both of which show good experimental performance.
- Has mathematical rigor in both breaking down the loss function, as well as in the theoretical analysis that follows.
- Compares against multiple state-of-the-art MVC methods and shows superior performance on the chosen datasets.
- Conducts a wide range of experiments, including ablation studies and parameter sensitivity analysis.

**Weaknesses:**

- The presentation of the mathematical parts of the preliminaries, discussion and analysis is quite opaque. Variables and terms are often explained only after they have already been used in equations. The equations tend to be quite cluttered and difficult to parse.
- The datasets chosen are less than ideal, and are mostly derivative from single-view datasets. There are other multi-view datasets such as NUS-WIDE and 3-Source News which are potential candidates for publicly available MV datasets.
- The assumptions (and thus applicability) of the methods are restrictive. I.e. assuming mutual redundancy across all views. As they mention in the supplementary, their methods perform poorly on heterogeneous MV data, limiting their applicability.
- The paper was not self-contained; the supplementary material was essentially required to understand the a lot of the details. For eg. the limitations of the methods were only mentioned in the supplementary material. The supplementary material must not be used as additional pages, and the paper itself should be able to stand alone.

**Questions:**

Overall questions:

- This paper is more about representation learning for clustering than clustering itself, right? Similar to spectral clustering. Not that that's a bad thing, it's just a little misleading. You spend a lot of the paper talking about MVC but the actual clustering is just k-means on top of a learned representation. It might help to be clear about this up front.
- As I mentioned in the second point of the Limitations, it seems like this method is also weak to only partial information existing in views. How would you remedy this? Would ensemble variational methods work here as well?
- For the MNIST experiments, I am confused about your choice to use pairs of the same digit (but written by different people) as two views. The data distribution of both the views are very similar, then. Also, if you pair up digits, shouldn't your dataset size be 35000 (halved) and not 70000? Or are you using both (A, B) and (B, A) for each pair? If so, this doesn't really seem like a good multi-view dataset since the views themselves are basically indistinguishable overall.
- In table 3, the 3rd row seems to not have any useful information. It's clear that $L_{rec}$ would be required since you're using an auto-encoder. Instead, maybe you can have $L_{rec} + L_{suf}$ which you don't have here. That would be interesting to see.


I will leave line-by-line comments here (since there doesn't seem to be a better place for this). There are a few typos here and there, but I won't bother too much with those:

- [Line 32] Did you mean maximizing MI between representation* and output (not input and output)
- [Line 82] "... to quantify amount of" -- sentence is not complete.
- [Section 3] You use y, z without explaining/defining what they are. Clustering is an unsupervised task, so what does this mean in this context? I'm guessing it means cluster assignment but it isn't clear.
- [Line 164] I may be wrong but isn't $\phi^i$ the generational parameter set, and not $\theta^i$?
- [Line 180] What is a "pseudo-label?" A cluster assignment? You should define this earlier.
- [Line 164] What do you mean by unique distribution?
- [Line 186] This equation is very hard to parse. For cleaner appearance, you should consider using \left( and \right) to have larger parantheses. You could also remove the superscripts just for these long equations where there is no $j$, and just leave a note below. Also, shouldn't there be a conditioning on y in the first term? Lastly, $KL$ -> $D_{KL}$.
- [Line 198] maximizing*
- [Equation 9] There should be no expectation here, right? The $D_{KL}$ absorbs it, I think. Also, for the second term. the first distribution uses $z^j$ instead of $z^i$.
- [Line 243] Should it be $z^j_m$ here? Also, what is $P^{\otimes n}$?

**Limitations:**

- [Author mentioned] Heterogeneity in the data (eg. in terms of dimensions of features) affects performance significantly.
- Restrictive assumptions are made on mutual redundancy between views. Multi-view data often has only partial information available in each view. I.e. you might need more than one view to get the complete picture. This also seems like a weakness of the methods.

---

> ### Author Rebuttal · Authors · 2023-08-09
>
> We appreciate the reviewer for the thoughtful comments and feedback. Below please find our detailed responses to the questions.
>
> 1.	This paper is more about representation learning for clustering than clustering itself, right?
>
> We appreciate your insightful observation. You're correct in noting that our paper primarily focuses on the aspect of representation learning for clustering, which is analogous to spectral clustering. The use of the k-means method on top of the learned representation is indeed a subsequent step, and the Multi-View Representation Learning (MVRL) method forms the core of our discussion.
> In response to your feedback, we will revise our manuscript to make this point more explicit upfront. We believe this adjustment better frames the main contributions of our work and minimizes any potential misunderstandings about the paper's focus.
> Thank you again for pointing out this nuance.
>
> 2. As I mentioned in the second point of the Limitations, it seems like this method is also weak to only partial information existing in views. How would you remedy this? Would ensemble variational methods work here as well?
>
> Thank you for your insightful comments. We appreciate your mention of the issue of partial information existing in views, which indeed is a challenge that our method currently faces. The original intention of MVC is that a single view only contains partial information, and it is difficult to depict the complete clustering structure. MVC combines complementary information from multiple views to achieve better clustering results. If "partial information" refers to missing data, one possible strategy could involve integrating additional data preprocessing steps, such as imputation methods for handling missing data, which could potentially enhance the robustness of our method when dealing with partial views.
>
> Regarding your suggestion of using ensemble variational methods, we believe it's a very promising direction. Ensemble methods could indeed provide a solution to this issue by leveraging the consensus among multiple models, each trained on a different subset of the data. This approach could help to address the inherent uncertainty and variability in the data, and thus potentially improve the performance of our method when dealing with partial views.
> We will follow your suggestion to add a discussion about these potential strategies to the manuscript. We believe that this addition will stimulate further research on this topic and provide a roadmap for improving the current limitations of our method.
> We hope this adequately addresses your concerns. We greatly value your suggestions and look forward to any further feedback you may have.
>
> 3. For the MNIST experiments, I am confused about your choice to use pairs of the same digit (but written by different people) as two views. The data distribution of both the views are very similar, then. Also, if you pair up digits, shouldn't your dataset size be 35000 (halved) and not 70000? Or are you using both (A, B) and (B, A) for each pair? If so, this doesn't really seem like a good multi-view dataset since the views themselves are basically indistinguishable overall.
>
> Thank you for your thoughtful feedback, which will undoubtedly improve the quality of our paper.  The MNIST-MV data we used in the experiments is a public available dataset and has been widely used in previous multi-view studies, e.g., "Deep safe multi-view clustering: Reducing the risk of clustering performance degradation caused by view increase,  CVPR 2022", "Multi-VAE: Learning Disentangled View-common and View-peculiar Visual Representations for Multi-view Clustering, ICCV 2021", and "Multi-view Semantic Consistency based Information Bottleneck for Clustering, arxiv 2023".
>
> Regarding our choice to use pairs of the same digit written by different people as two views, the rationale behind this decision was to examine the ability of our model to identify and learn from subtle differences in similar-looking data. While the two views may seem indistinguishable at a macro level, they can contain minute differences at a micro level. These differences are due to the unique handwriting styles of different individuals, which our model aims to capture and learn from.
>
> As for your question about the dataset size, you are correct in your understanding. We indeed used both (A, B) and (B, A') pairs (A' may not be equal to A), effectively maintaining the original dataset size of 70,000. The reason behind this approach was to increase the diversity of our training data and further test the robustness of our model. However, we understand your concern about the potential impact on the multi-view nature of the dataset.
>
> In light of your comments, we will ensure that our choices are properly justified in the revised manuscript.
>
>
> 4. In table 3, the 3rd row seems to not have any useful information. It's clear that $L_{rec}$ would be required since you're using an auto-encoder. Instead, maybe you can have  $L_{rec} +L_{suf} $ which you don't have here. That would be interesting to see.
>
>
> Thank you for your insightful comments.  We will remove the 3rd row to avoid any confusion and the results of the ablation experiment with the inclusion of $L_{rec} + L_{suf}$ can be found in the table below.
>
> |Dataset | Multi-MNIST | Multi-Fasion|Multi-COIL-20 |
> |---|---|---|---|
> | ACC|  98.4| 84.6 |86.9|
> | NMI    |  96.0| 80.8 |91.0|
> | ARI |   96.6| 75.2 |83.1|
>
> We found that model with only the $L_{rec} + L_{suf}$ terms  perform worse than SUMVC. This is because $L_{suf}$ helps the model learn the distributional features of the latent layer. Therefore, the lack of this constraint makes it challenging for the model to effectively learn these features.   We hope these adjustments and explanations address your concerns satisfactorily.

---

> > ### Comment · Reviewer_NC94 · 2023-08-17
> >
> > I acknowledge the author's comments, and have gone through the other reviewers' feedback.
> >
> > The authors' responses have answered my questions well. I believe that the presentation of the paper will be much clearer after incorporating the changes they mentioned in their rebuttal. I hope that weakness #4 above will also be addressed in their changes to the manuscript (i.e. paper not being self-contained and needing the appendix to really understand it).
> >
> > My main remaining concern is that the evaluations/experiments are not conducted on natural multi-view datasets, and rather on modified single-view datasets. While I understand that MNIST-MV is commonly used in literature, lacking experimental evaluations on natural multi-view datasets detracts from the impact of the contribution. It seems that most of the reviewers have similar concerns on the experimental evaluations.
> >
> > If additional experiments on other datasets have been conducted since then, I would like to see those evaluations as well.
> >
> > At this point, I intend to keep my original score.

---

> > > ### Author Response · Authors · 2023-08-19
> > >
> > > Dear Reviewer NC94,
> > >
> > > We appreciate your response to our rebuttal and the additional questions raised. Regarding the evaluations being conducted on modified single-view datasets rather than natural multi-view datasets, we’d like to clarify that for the MULTI-COIL-10 and MULTI-COIL -20 datasets, they are not derived from single-view data, and each view represents different angles at which the photos were taken.
> > >
> > > To further address your concerns, we have conducted additional experiments using two widely used multi-view datasets: the REU dataset and the HW dataset. We have also introduced five additional comparison methods, namely IDEC (Improved Deep Embedded Clustering), CSMSC (Consistent and Specific Multi-View Subspace Clustering), FMR (Flexible Multi-View Representation Learning for Subspace Clustering), GMC (Graph-Based Multi-View Clustering), and CGMSC (Multi-View Subspace Clustering with Adaptive Locally Consistent Graph Regularization). The results are presented in the tables below.
> > >
> > > | Method (REU)| IDEC| SAMVC | GMC | DEMVC | SUMVC |
> > > | --- | --- | --- | --- | ---  | ---  |
> > > | ACC | 46.0|18.8  | 19.8|46.7|58.3 |
> > > | NMI| 25.2|4.6    |  13.8| 25.3 |60.0|
> > > | ARI | 18.0|  0.3|   1.3| 20.4|47.5|
> > >
> > > | Method (HW)| SAMVC | FMR |CSMSC | CGMSC | FMVACC | SUMVC |
> > > | --- | --- | --- | --- | ---  | ---  | --- |
> > > | ACC | 76.4  | 86.1| 89.8 |69.1|89.5| 96.4 |
> > > | NMI| 84.4   |   76.5|83.0 |81.8|86.0|93.2|
> > > | ARI |  73.9|   72.6| 79.5|69.5|85.0|92.4|
> > >
> > >
> > >
> > >
> > > It can be observed that our method performs well on these datasets. We will add new comparative methods and datasets in the final version. We hope our response has addressed your concerns.

---

> > > > ### Comment · Reviewer_NC94 · 2023-08-20
> > > >
> > > > Thank you for your reply! It looks my main concerns have all been addressed. I have raised my score.

---

### Official Review · Reviewer_79CP · 2023-07-06

**Soundness:** 3 good
**Presentation:** 3 good
**Contribution:** 3 good
**Rating:** 6
**Confidence:** 5

**Summary:**

A consistent variational lower bound is provided to explore the consistent information among views for multi-view clustering, based on which SCMVC (simple consistent multi-view clustering) is proposed. To enhance consistent information and minimize unnecessary information among views, a sufficient representation lower bound is further proposed.

**Strengths:**

(1) The main idea is novel, necessary preliminary knowledge and sufficient theoretical analysis is given.

(2) Experiments conducted on real multi-view data demonstrate the good performance of proposed methods. The codes are available.

(3) The findings of this study have the potential to contribute to the advancement of multi-view clustering techniques.

**Weaknesses:**

(1) It is said the proposed model does not perform well on datasets with strong heterogeneity between views, but no evidence or experiments support this statement. How does the heterogeneity affect the performance should be explained.

(2) The paper shows quantitative results of the proposed SUMVC and SCMVC. However, I expect some visualization results to show the difference between these two methods.

(3) A thorough analysis of potential drawbacks, and practical considerations would enhance the overall strength of the paper.

(4) The tested data sets in this paper contain a small number (<=3) of views. It is suggested to add data sets with more than three views for discussion.

(5) The writing should be further improved.

**Questions:**

Please see ‘Weaknesses’.

**Limitations:**

Yes

---

> ### Author Rebuttal · Authors · 2023-08-09
>
>  We thank the reviewer for the thoughtful comments and feedback. Below, we provide our responses to the key questions that were raised by the reviewer.
>
> 1. It is said the proposed model does not perform well on datasets with strong heterogeneity between views, but no evidence or experiments support this statement. How does the heterogeneity affect the performance should be explained.
>
> Thank you for your insightful comment. We have conducted additional experiments on the BDGP dataset  to substantiate our initial statement. BDGP contains 5 different types of drosophila. Each sample has visual and textual views. It has high heterogeneity between views. To mitigate the heterogeneity issue, we explored the use of a Multilayer Perceptron (MLP) to harmonize dimensions. We also tested the efficacy of sharing parameters between views. Both strategies have shown promise in alleviating the performance decrement in heterogeneous datasets. We  made a comparison between four models: (1) SCMVC-NS (not-share): the original SCMVC  without consideration of heterogeneity between views, (2) SCMVC-MLP: the improved  SCMVC  using a single layer MLP as a consideration of unifying dimensions and sharing parameters between views, (3) SUMVC-NS: the original SUMVC without consideration of heterogeneity between views, and (4) SUMVC-MLP: the improved SUMVC using a single layer MLP as a consideration of unifying dimensions and sharing parameters between views.
>
>  | Method | SCMVC-NS | SCMVC-MLP | SUMVC-NS| SUMVC-MLP |SAMVC |FMVACC |
> | --- | --- | --- | --- | ---  | ---  | --- |
> | ACC | 49.9   |  55.8| 55.3 |71.4|51.3|58.6|
> | NMI| 44.8    |   48.6| 39.9 |60.3|45.2|36.8|
> | ARI |   29.6|   35.1| 27.3|45.3|19.6|44.3|
>
> The experimental outcomes demonstrate that while our model does not reach the ideal level on the BDGP dataset with strong heterogeneity between views, it delivers a comparable performance to the existing models, i.e.,  (Fast multi-view anchor-correspondence clustering, NeurIPS 2022) and SAMVC (Self-paced and auto-weighted multi-view clustering, NC 2020).
> We believe these additional experiments and proposed solutions provide a clearer understanding of our model's behavior under varying degrees of heterogeneity. We hope this addresses your concerns satisfactorily and provides a more comprehensive view of our work.
>
>
> 2. The paper shows quantitative results of the proposed SUMVC and SCMVC. However, I expect some visualization results to show the difference between these two methods.
>
> Thank you for your insightful suggestion. We agree that providing visual results could further illustrate the differences between the SUMVC and SCMVC methods, in addition to the quantitative results we have already provided.
> To this end, we will provide additional visualizations to better elucidate the distinctions between the two methods.
>
> 3.  A thorough analysis of potential drawbacks, and practical considerations would enhance the overall strength of the paper.
>
> Thank you for your constructive feedback. We discussed potential drawbacks in the Appendix:
> our model does not perform well on datasets with particularly strong heterogeneity between views, such as huge differences in dimensions of different views.
> We will add a new paragraph in the Conclusion section to address this issue. In this section, we critically analyze the limitations of our approach, discuss possible scenarios where our method may not perform optimally, and outline practical considerations for researchers and practitioners intending to adopt our methodology. Please see the details below:
>
> The heterogeneity between views (i.e., the distributions of different views are obviously different) can make it more difficult for the VAE to learn a meaningful latent representation of the data. When the views are highly dissimilar such as views of BDGP, it may be challenging for the VAE to find a shared low-dimensional representation that captures the important features of both views. This can lead to suboptimal performance and poor reconstruction quality.
>
> We believe that this addition will provide the reader with a more comprehensive understanding of our work, and more importantly, it will stimulate further research to overcome the highlighted limitations.
>
> For practical considerations, multi-view clustering has broad prospects in the fields of data analysis and pattern recognition. The main idea behind multi-view clustering is to utilize multiple data sources or feature sets to perform clustering analysis, thereby improving the accuracy and robustness of clustering results.
> Multi-view clustering can be applied in various domains such as bioinformatics, social network analysis, image processing, and text mining. In bioinformatics, multi-view clustering can be used for the analysis of gene expression data, combining different experimental platforms and data types to discover more accurate gene expression patterns and biological features. In social network analysis, multi-view clustering can integrate users' social relationships, interest tags, and behavioral data to achieve more refined user segmentation and community discovery.
>
>
> 4. The writing should be further improved.
>
> We appreciate your feedback on our writing. We will review and revise our manuscript. We focus on improving the clarity, coherence, and conciseness of our writing, ensuring our arguments are well-structured, and that our language is precise.

---

> > ### Comment · Reviewer_79CP · 2023-08-22
> > **I**
> >
> > Dear authors，
> >   Thanks for your detailed reply. My concerns have been addressed by your response. The addtional emperiments comparsions are hoped to  add in the revised version. I would like to increse my scores into 'weak accept'.

---

### Decision · Program_Chairs · 2023-09-21

**Decision:**

Accept (poster)

**Comment:**

This well-written paper has been reviewed by five knowledgeable reviewers who all recommended its acceptance (three straight accepts and two weak accepts). The authors have engaged the reviewers in productive discussions which addressed the key concerns raised by the reviewers. The presented work involves multi-view clustering and it will be of interest to the NeurIPS audience.